# An Oxylipin-Related Nutrient Pattern and Risk of Type 1 Diabetes in the Diabetes Autoimmunity Study in the Young (DAISY)

**DOI:** 10.3390/nu15040945

**Published:** 2023-02-14

**Authors:** Teresa Buckner, Randi K. Johnson, Lauren A. Vanderlinden, Patrick M. Carry, Alex Romero, Suna Onengut-Gumuscu, Wei-Min Chen, Oliver Fiehn, Brigitte I. Frohnert, Tessa Crume, Wei Perng, Katerina Kechris, Marian Rewers, Jill M. Norris

**Affiliations:** 1Department of Epidemiology, Colorado School of Public Health, CU Anschutz, Anschutz Medical Campus, Aurora, CO 80045, USA; 2Department of Kinesiology, Nutrition, and Dietetics, University of Northern Colorado, Greeley, CO 80639, USA; 3Department of Biomedical Informatics, CU School of Medicine, Anschutz Medical Campus, Aurora, CO 80045, USA; 4Colorado Program for Musculoskeletal Research, Department of Orthopedics, CU School of Medicine, Anschutz Medical Campus, Aurora, CO 80045, USA; 5Health Center for Public Health Genomics, University of Virginia, Charlottesville, VA 22903, USA; 6NIH-West Coast Metabolomics Center, University of California-Davis, Davis, CA 95616, USA

**Keywords:** oxylipin, nutrient pattern, type 1 diabetes, reduced rank regression

## Abstract

Oxylipins, pro-inflammatory and pro-resolving lipid mediators, are associated with the risk of type 1 diabetes (T1D) and may be influenced by diet. This study aimed to develop a nutrient pattern related to oxylipin profiles and test their associations with the risk of T1D among youth. The nutrient patterns were developed with a reduced rank regression in a nested case-control study (*n* = 335) within the Diabetes Autoimmunity Study in the Young (DAISY), a longitudinal cohort of children at risk of T1D. The oxylipin profiles (adjusted for genetic predictors) were the response variables. The nutrient patterns were tested in the case-control study (*n* = 69 T1D cases, 69 controls), then validated in the DAISY cohort using a joint Cox proportional hazards model (*n* = 1933, including 81 T1D cases). The first nutrient pattern (NP1) was characterized by low beta cryptoxanthin, flavanone, vitamin C, total sugars and iron, and high lycopene, anthocyanidins, linoleic acid and sodium. After adjusting for T1D family history, the HLA genotype, sex and race/ethnicity, NP1 was associated with a lower risk of T1D in the nested case-control study (OR: 0.44, *p* = 0.0126). NP1 was not associated with the risk of T1D (HR: 0.54, *p*-value = 0.1829) in the full DAISY cohort. Future studies are needed to confirm the nested case-control findings and investigate the modifiable factors for oxylipins.

## 1. Introduction

Type 1 diabetes (T1D) is an autoimmune disease that results from the destruction of the insulin-producing pancreatic beta cells, ultimately leading to dependence on insulin. Inflammation of the pancreatic islet cells, involving both T cells and macrophages, is a key characteristic of T1D [1,2]. A variety of the potential environmental factors have been investigated, but the etiology is yet to be fully elucidated. Pancreatic islet inflammation may be initiated by an environmental trigger, for example, infections, including enteroviruses [3,4,5], altered gut permeability [6] and microbiome changes, have been proposed [7].

The innate immune system plays a role in the pathogenesis of T1D through increasing the islet cell inflammation and pro-inflammatory cytokines and chemokines [8]. The inflammatory cytokines CXCL10 [9] and interleukin-1β (IL-1β) [10,11] may lead to damage of the islet cells in the pancreas, and individuals that develop T1D may have a reduced ability to create IL-4 prior to T1D development [12]. There were differences in IFN-γ and IL-6 between those that did and did not progress from islet autoimmunity (IA) to T1D in the Diabetes Autoimmunity Study in the Young (DAISY) [13]. In support of the relationship between inflammation and T1D, we found that plasma oxylipins, bioactive lipids generated by the oxidation of polyunsaturated fatty acids (PUFA) with both pro-inflammatory and pro-resolving inflammation actions, are associated with T1D [14]. Oxylipins are lipid mediators derived from omega-3 (n3) and omega-6 (n6) PUFA. In DAISY, an oxylipin profile characterized by linoleic acid (LA)- and alpha-linolenic acid (ALA)-related compounds was associated with a lower risk of T1D. On the other hand, an oxylipin profile characterized by the inflammatory arachidonic acid (ARA)-related compounds was associated with a higher risk of T1D.

Diet plays a role in regulating the inflammatory conditions within the body. For example, we found that the intake of the precursor FA is associated with the plasma levels of some oxylipins in DAISY [15]. While it may not be possible to prevent inflammation-causing insults, it may be possible to identify the modifiable interventions that reduce the inflammatory response to these exposures. Anti-inflammatory nutrients, including n3 fatty acids (FA), are associated with a reduced risk of IA [16,17,18]. The supplementation of vitamin D in children, which has both immunomodulatory and anti-inflammatory properties, has been shown to be protective of T1D [19]. In a case report, high-dose vitamin D combined with n3 FA have slowed beta cell destruction [20]. On the other hand, serum levels of the FA palmitoleic acid, a pro-inflammatory nutrient, is associated with the development of IA [21]. In DAISY, a higher total sugars intake and glycemic index—also associated with inflammation [22,23,24]—increased the risk of progression from IA to T1D [25,26].

We hypothesized that both genetic and dietary factors influenced oxylipin levels, so we adjusted for the genetic variants associated with oxylipins prior to developing the dietary pattern. The aim of this study was to develop an oxylipin-related nutrient pattern that was independent of genetics and test the relationship between the nutrient pattern and the risk of T1D in a cohort of children at risk of T1D. We used nutrients, rather than foods, to develop the intake pattern because nutrients may be more portable to different cultures and dietary practices than food. To do this, we used genetically adjusted oxylipin profiles that are associated with T1D [14] as the response variables for a reduced rank regression (RRR) model and tested the resulting nutrient patterns with the risk of T1D in the DAISY cohort.

## 2. Materials and Methods

### 2.1. Study Population

DAISY is a prospective cohort study of 2547 children at risk of T1D. The participants came from two groups at a high risk of developing T1D: the first degree relatives of patients with T1D (*n* = 1123) and the infants screened at birth for genetic risk (*n* = 1424). Of the infants born at St. Joseph’s Hospital in Denver, CO, USA, whose cord blood was screened for the HLA diabetes-susceptibility alleles [27], those with high-risk T1D genotypes were invited to participate. These HLA genotypes were defined as DRB1*04, DQB1*0302/DRB1*0301, DQB1*0201 (DR3/4 DQ8) [27]. The participants were followed prospectively with study visits at 9, 15 and 24 months of age and annually thereafter.

### 2.2. IA and T1D Case Definition

As described previously, the participants were tested for IA using radio-binding immunoassays for the serum autoantibodies to GAD65, IAA, IA-2 and ZnT8 [28]. For this study, IA was defined as testing positive for at least one islet autoantibody on two or more consecutive visits or testing positive for islet autoantibodies followed by a diagnosis of T1D at the next visit (within a year) [29]. T1D was diagnosed by standard criteria [30].

### 2.3. Nested Case-Control Study

An IA nested case-control study was selected from within the DAISY. IA controls (*n* = 207) who were autoantibody-free and T1D-free at the time of matching, were frequency matched to 207 IA cases on age at the IA seroconversion of the case, race/ethnicity and sample availability. A T1D nested case-control study was also selected and consisted of the 89 IA cases that went on to develop T1D and their 89 controls. A maximum of four samples were selected from visits defined as the following.

The earliest sample (at 9–15 months)The sample collected just prior to seroconversion (or age-matched visit for the IA controls)The sample collected just after seroconversion (i.e., the sample in which autoantibodies were first detected)The sample collected just prior to T1D diagnosis (for the T1D case-control study)

### 2.4. Measurement of Oxylipins

Oxylipins were quantified as described by Pedersen et al. [31]. The extracted oxylipins were separated and quantified using a Waters i-Class Acquity UHPLC system coupled with a Sciex 6500+ QTRAP mass spectrometer operated in a negative ionization mode. Oxylipins were quantified by targeted, retention-time specific, multiple reaction monitoring (MRM) ion transitions. There were quantified values for 24 n6-related oxylipins, including 14 ARA-derived and 10 LA-derived oxylipins. There were 12 measured n3-related oxylipins, which included six ALA-derived, four Docosahexaenoic acid (DHA)-derived and two Eicosapentaenoic acid (EPA)-derived oxylipins. The oxylipins were Box-Cox transformed [32] using the forecast package in R (v8.1, R package version 8.12 [33]) and considered normally distributed for further analyses [34].

### 2.5. Genotyping

The DNA samples were genotyped on the custom designed TEDDY-T1D Exome array at the University of Virginia (UVA) Genome Sciences Laboratory following the manufacturer’s protocol (Illumina). Additional variants were imputed using the Trans-Omics for Precision Medicine (TOPMed) multi-ancestry reference panel (Version r2), resulting in a total of 292,174,934 variants [35,36,37]. The Rsq was calculated as the estimated squared correlation of the imputed genotype and the true (unobserved) genotype. We selected SNPs with a minor allele frequency ≥ 0.01 and an Rsq ≥ 0.7, which led to the inclusion of 9,222,144 variants.

### 2.6. Dietary Assessment

Diet was measured using a Willett 111-item semi-quantitative food frequency questionnaire (FFQ) in the DAISY cohort [38,39]. The FFQ was administered annually to the parents, inquiring on the previous year’s dietary intake of the child starting at 2 years of age, and was validated for use in the DAISY population [40]. After the age of 10, the children recalled their own diets by completing the youth/adolescent questionnaire (YAQ) [41,42]. Both questionnaires produce similar estimates of the nutrients and may be combined for analyses, as reported [42]. As compared to a 3-day food record, the FFQ produced similar dietary patterns with a modest agreement in adolescents [43]. The nutrient values were residually adjusted for the total energy intake [44].

### 2.7. Overview of Statistical Analysis

To date, most studies on diet and T1D have examined single nutrients or food items. However, foods and nutrients are generally eaten in combination and may interact synergistically or antagonistically. Testing combinations of foods or nutrients through comprehensive patterns of intake may improve our understanding of how diet may promote or mitigate disease risk [45,46,47,48]. RRR is a data-driven approach for creating dietary patterns by creating linear combinations of predictors that explain the maximum covariance of a set of response variables [49]. RRR is useful for studying disease pathophysiology because it utilizes *a priori* knowledge of disease-specific markers and *a posteriori* information on dietary intake for creating dietary patterns for a disease outcome. RRR may be useful in leveraging T1D-specific inflammatory markers in determining a relevant dietary pattern. 

We used RRR to create the nutrient patterns, and this statistical method did not allow for the use of longitudinal data in identifying the optimal set of predictors (nutrients) for the outcome (oxylipin PC1 and PC2). So, we created the summary measures of oxylipins and the nutrients in order to create the nutrient pattern. We employed a five-step process in this analysis (Figure 1). 

We derived the genetically adjusted oxylipin PCs in the nested IA case-control study.We derived the average nutrient measures in the nested IA case-control study.We developed the genetically adjusted nutrient patterns in the nested IA case-control study using RRR.We tested the associations of the nutrient patterns with incident T1D risk in the nested T1D case-control study.We examined the validity of the findings in the full DAISY cohort.

ANOVA was used to describe the baseline characteristics of the study population. The statistical analyses (except for the genome-wide association study (GWAS)) were conducted using SAS v.9.4.

(1)
*Derivation of Genetically Adjusted Oxylipin PCs.*


We first created the response variable for the RRR, namely the genetically adjusted oxylipin PCs. In order to maximize the sample size, we included everyone with both exome chip and oxylipin data, which had been measured in the participants in the IA nested case control study (*n* = 339, Figure 1). We had multiple measures of oxylipins, so we created a summary measure of each oxylipin. To do this, we used a linear mixed model with a random intercept and an unstructured covariance structure with age as the predictor and the oxylipin as the outcome, as previously described [14]. The subject-specific intercept from these models was used as an age-adjusted measure of the average oxylipin level over time. Since oxylipins share precursor FAs and enzymes, we used principal components analysis (PCA) of the oxylipin intercepts to measure the oxylipin profile, as described [14]. Based on the screen plot (Appendix A) of the oxylipins that loaded onto the PCs, we selected PC1 and PC2 to reflect the distinct oxylipin profiles in the DAISY children, as described [14]. PC1 represented LA- and ALA-related oxylipins and PC2 represented ARA-related oxylipins (Appendix A). We previously found that these oxylipin profiles were associated with risk of T1D in DAISY [14].

Since we hypothesized that both genetic and environmental factors (e.g., diet) impact oxylipins, and it is recommended to adjust the response variables before the implementation of the RRR [50], we adjusted for genetic predictors of the oxylipin profiles. We conducted a GWAS (manuscript under review) with oxylipin PC1 and PC2 as the outcomes and adjusted it for sex and ancestry using PLINK v1.9 [51]. SNPs that met a suggestive significance of *p* < 5 × 10^−5^ for oxylipin PC1 or PC2 were included in a stepwise linear regression model for oxylipin PC1 or PC2. SNPs were removed from the model if *p* > 5 × 10^−5^, and the model with the highest adjusted R^2^ was selected. Oxylipin PC1 and PC2 were then residually adjusted for the selected variants (Appendix A).

(2)
*Derivation of Average Nutrient Measures*


To create the nutrient measures for the RRR, we included all the participants with complete data (exome chip, oxylipin and measured FFQ). Four participants from the GWAS in Step 1 did not have the measured FFQ data, so *n* = 335 participants were included (Figure 1). Nutrient intake was used for the dietary variables because nutrients may be more portable across cultures and dietary practices compared to specific food items. Additionally, because RRR assumes a Gaussian distribution of variables [52], nutrients, which are aggregated across foods, are more likely to follow a Gaussian distribution compared to the measurements of food intake from a FFQ. We selected macro- and micro-nutrients and removed the nutrients that were components of one another (e.g., total sugar was selected, therefore fructose, lactose, glucose, etc. were not included) for the ease of interpretation, leading to a list of 48 nutrients (Appendix A). We summarized multiple measures of the nutrient intake. We used a linear mixed model with a random intercept and unstructured covariance structure with age as the predictor and nutrient intake (adjusted for the total energy intake (TEI) using the residual method [53]) as the outcome. The subject-specific intercept from these models was used as an age-adjusted measure of the average nutrient intake over time and was standardized to a mean of 0 and a standard deviation of 1.

(3)
*Nutrient Pattern Development.*


We used RRR to create the nutrient patterns related to the biomarkers of inflammation in T1D. To maximize the sample size, we used all the participants with measured genetic, oxylipin and dietary intake data (*n* = 335, Figure 1). Genetically adjusted oxylipin PC1 and PC2 were used as the response variables. We selected all the nutrients (using the age-adjusted average intake measures) that were associated with genetically adjusted oxylipin PC1 or PC2 (*p* < 0.2), using a linear regression to be entered in the RRR model [54,55,56,57,58,59]. All the nutrients associated with genetically adjusted oxylipin PC1 and/or PC2 were used as the diet variables, and both genetically adjusted oxylipin PC1 and PC2 were used as the response variables for the RRR model. The nutrient patterns that best explained the oxylipin variation were chosen by minimizing the predicted residual sum of squares (PRESS). The nutrient loadings were used to interpret the nutrient patterns, and the nutrients with loadings >|0.2| were of interest. 

(4)
*Testing the association in the nested case-control study.*


We tested the nutrient pattern in a logistic regression model with the risk of T1D in the T1D nested case-control study (*n* = 69 T1D cases, 69 controls), adjusted for family history of T1D, the high-risk HLA genotype, sex and race/ethnicity. We also tested the relationship between the individual nutrients selected for the nutrient pattern and the risk of T1D to determine whether the association between the nutrient patterns and T1D was being driven by a particular nutrient within the pattern.

(5)
*Nutrient patterns and risk of T1D longitudinally within the full DAISY cohort.*


We tested the association between the nutrient pattern and the risk of T1D longitudinally in the full DAISY cohort using a joint Cox proportional hazards (PH) model. We included everyone with at least one measured FFQ, *n* = 1933, including 81 T1D cases (which also included the 69 T1D cases included in Step 4), (Figure 1). Joint models allow for analyzing the repeated measurement of the exposure data with time-to-event outcomes, and it utilized the extensive dietary intake of the data available through DAISY. Joint models estimate the exposure variable profile over time, assume a smooth change over time and can accommodate the missing data or measurements taken at unequal intervals [53]. For the joint model, the age of the FFQ measurement was used to estimate the trajectory of the nutrient pattern. 

We adjusted the nutrient intake for the TEI using the residual method [44] for all the FFQs in the full cohort and then standardized the nutrient intake to a mean of 0 and standard deviation of 1. We applied the nutrient patterns to all the FFQ measures by scoring the nutrients calculated from each food record according to the weights derived from the nutrient pattern [60]. A higher score indicated an intake closer to the nutrient pattern. We determined that a quadratic trajectory with five knots was the best fit for the data using AIC to assess the model fit. The nutrient pattern was tested as an average (intercept) and as the cumulative effect. When applying an RRR dietary pattern to another population, it is recommended to implement a “simplified pattern” that consists only of foods or nutrients that load as important [50]. Therefore, we scored a “simplified nutrient pattern,” with only nutrients with loading values >|0.2|. The joint models were adjusted for family history of T1D, the high-risk HLA genotype, race/ethnicity and sex.

## 3. Results

### 3.1. T1D Nested Case-Control Characteristics

The participants in the T1D nested case-control study (*n* = 69 T1D cases, 69 T1D controls) are described in Table 1. The T1D cases were more likely to have the high-risk HLA genotype (*p* = 0.0002). In the T1D nested case-control study, 44.9% were female and 89.8% of the participants reported non-Hispanic white ethnicity, but there were no significant differences in sex, race/ethnicity, or family history of T1D between the T1D cases and controls.

### 3.2. Development of Oxylipin Patterns

We identified 13 SNPs that together explained 55.83% of the variation in oxylipin PC1 (Appendix A). We identified 12 SNPs that together explained 55.88% of the variation in oxylipin PC2 (Appendix A). In the subsequent analyses, we adjusted the oxylipin PCs for these SNPs to derive the genetically adjusted oxylipin PCs.

### 3.3. Development of Nutrient Pattern

Next, we considered 10 nutrients that were associated with genetically adjusted oxylipin PC1 and six nutrients associated with genetically adjusted PC2 (*p* < 0.2) (Table 2) using all the participants with measured genetics, oxylipins and diet (*n* = 335, Figure 1). The nutrients most significantly associated with the oxylipin PCs were sodium, which was positively associated with genetically adjusted PC1 (β estimate: 0.113, *p* = 0.0014), and beta cryptoxanthin (β estimate: −0.106, *p* = 0.0026) and flavanone (β estimate −0.086, *p* = 0.0148), which were both negatively associated with genetically adjusted PC1. We included 14 nutrients for the RRR model (LA and vitamin C intake were associated with both genetically adjusted oxylipin PC1 and PC2), with genetically adjusted oxylipin PC1 and PC2 as the response variables for the model.

We extracted two nutrient patterns using RRR. Nutrient pattern 1 (NP1) explained 9.5% of the variation in genetically adjusted PC1 and 0.01% of the variation in genetically adjusted PC2. The positive loadings indicated higher intakes, whereas the negative loadings indicated lower intakes. We considered the nutrients with loadings > |0.2| to be important based on the distribution of the factor loadings [61,62,63]. NP1 represented a diet low in beta cryptoxanthin, flavanone, vitamin C, total sugars and iron, and high in lycopene, anthocyanidins, LA and sodium (Figure 2, Appendix A). Nutrient pattern 2 (NP2) explained 9.5% of the variation in genetically adjusted oxylipin PC1 and 4.7% of the variation in genetically adjusted oxylipin PC2. NP2 represented a diet low in beta cryptoxanthin, vitamin C, potassium, flavonols, magnesium, vitamin B12 and LA. 

### 3.4. Nested Case-Control Association with T1D

NP1 and NP2 were tested in the nested T1D case-control study (*n* = 69 T1D cases, 69 controls), adjusted for family history of T1D, the high-risk HLA genotype, race/ethnicity and sex. NP1 was associated with a significantly lower risk of T1D (OR: 0.442, *p* = 0.0126) (Table 3). NP2 was not associated with T1D (OR: 0.590, *p* = 0.1362) (Table 3). We tested the component nutrients from NP1 and the risk of T1D (Appendix A) and none of the individual nutrients were significantly associated with T1D.

### 3.5. Longitudinal Association with T1D

Finally, we tested the nutrient pattern in the DAISY cohort, including people who did not have measured genetics or oxylipins, using a joint Cox PH model. This model estimated the trajectory of the nutrient pattern using age at the FFQ measurement and incorporated these estimates (either a cumulative value or an average value) with the standard error into the Cox PH model. Here, we included *n* = 1933 participants, including 81 T1D cases. The T1D cases were more likely to have the high-risk HLA genotype and a first degree relative with T1D and were more likely to report non-Hispanic white ethnicity (Table 4). We applied the original NP1 and the simplified NP1 to all the FFQ measurements and tested the original and simplified NP1 as the average and the cumulative value, adjusted for family history of T1D, the high-risk HLA genotype, race/ethnicity and sex. NP1 was not associated with T1D (Table 5).

## 4. Discussion

Using the data from 335 participants in the DAISY nested case-control study, we developed two nutrient patterns that each explained the variation in an oxylipin profile suspected to play a role in the pathophysiology of T1D via pro-inflammatory or pro-resolving pathways. One of these nutrient patterns (NP1) was inversely associated with the T1D outcome in our nested case-control study. This nutrient pattern was characterized by low beta cryptoxanthin, flavanone, vitamin C, total sugars and iron, and high lycopene, anthocyanidins, LA and sodium. While this nutrient pattern predicted a risk of T1D in the nested case-control in which it was developed, this association did not replicate in the full DAISY cohort.

### 4.1. Interpretation of the Findings for Nutrient Pattern 1 (NP1)

Both iron (loading value: −0.216) and vitamin C (loading value: −0.234) loaded negatively on NP1 (Appendix A). These results make sense, as it is well known that vitamin C increases iron absorption [64,65,66,67,68]. A diet high in both vitamin C and iron may lead to an overload of iron. Observationally, maternal iron may increase the risk of T1D [69], although the results are not consistent [70,71]. High dietary iron alongside dysregulation of pancreatic iron homeostasis may accelerate the onset of T1D in a mouse model [72]. Total sugars also loaded on to NP1 (−0.228), which aligns with similar findings that total sugars intake increased the risk of the progression from IA to T1D in the DAISY [26]. This is in line with the “overload hypothesis” that suggests that a high insulin load may trigger the onset of autoimmunity and T1D [73]. In this analysis, we were not able to assess the difference between heme and non-heme iron, or the timing of the iron and vitamin C intake, which may impact the absorption levels. Future studies on the interplay between iron intake, vitamin C and total sugars in the context of inflammation and T1D may reveal the mechanisms of these relationships in reducing the risk of T1D.

We also found that the antioxidants lycopene (loading value: 0.252) and anthocyanidins (loading value: 0.282) found in red and blue fruits and vegetables loaded positively on NP1, and, interestingly, the antioxidants beta cryptoxanthin (loading value: −0.441) and flavanone (loading value −0.360), generally found in yellow fruits and vegetables, loaded negatively. Anthocyanidins have been shown to reduce oxidative stress and autophagy in islet cells in vitro [74,75,76]. Blueberries, which have a high level of anthocyanins, decreased the production of pro-inflammatory ARA-related oxylipins post-exercise [77]. The antioxidant effect of anthocyanidins and lycopene may shift the oxylipin production to a pro-resolving, anti-inflammatory profile, reducing the risk of T1D. Sodium loaded the most strongly on NP1 (loading value: 0.475) and was significantly and positively associated with oxylipin PC1. In adults, salt loading increased the formation of LA-related oxylipins [78]. Additionally, sodium has been hypothesized to inactivate fatty acid desaturases that synthesize ARA from LA and EPA and DHA from ALA [79]. Sodium also increases the expression of the COX2 enzyme, which synthesizes oxylipins [80,81]. In NP1, sodium may play a role in shifting the oxylipin profile towards LA-related oxylipins, leading to the protective association of NP1.

NP1 was characteristic of a diet high in LA (loading value: 0.285), and also explained more variation in oxylipin PC1 compared to oxylipin PC2. Oxylipin PC1 represented LA- and ALA-related oxylipins and was associated with a decreased risk of T1D [14], which may explain the higher loading value for LA. In The Environmental Determinants of Diabetes in the Young (TEDDY) study, higher levels of LA in the erythrocyte membranes was associated with a reduced risk of IA in non-breastfed infants [82], suggesting a role of LA in the pathogenesis of T1D. LA has been linked to other inflammatory conditions. Mendelian randomization studies demonstrate that LA may reduce inflammation in asthma [83] and reduce the risk of autoimmune disorders [84].

Although LA and ARA are n6 FAs, which are generally associated with promoting inflammation, LA also plays a role in resolving inflammation [85]. LA reduces the mitochondrial damage inflicted through streptozotocin [86]. The difference between an LA- and ARA-related oxylipin profile may be related to resiliency to stressors through the promotion and resolution of inflammation, which has been proposed as a model for health. Oxylipins have been used as markers of this resiliency to stress [87]. LA-related oxylipins exhibit both pro-inflammatory and pro-resolving effects [85]. The protective effect of LA-related oxylipins may represent a child’s ability to respond to a stressor through the promotion and subsequent resolution of inflammation. ARA-related oxylipins, in contrast, do not demonstrate the same pro-resolution properties as LA-related oxylipins [88]. NP1 explained 9.5% of the variation in genetically adjusted PC1 and 0.01% of the variation of genetically adjusted PC2, so this nutrient pattern may be measuring a diet that promotes a state of resiliency to stress. This is a similar percent explained in other RRRs with biomarkers as response variables [89,90,91] and similar factor loadings to other RRR-derived dietary patterns [92,93,94].

### 4.2. Interpretation of Findings for Nutrient Pattern 2 (NP2)

NP2 demonstrated a protective association with T1D, although this was not significant (OR: 0.590, *p* = 0.1362). Similar to NP1, beta cryptoxanthin and vitamin C both loaded negatively. However, LA intake loaded negatively on this contrary to NP1. NP2 explained 4.7% of the variation in genetically adjusted oxylipin PC2, whereas NP1 explained 0.01% of the variation in genetically adjusted oxylipin PC2. As NP2 is related to both oxylipin PC1 and PC2, which had opposite associations with T1D [14], this may explain the non-significant findings.

### 4.3. Replication in the Full DAISY Cohort 

The protective association of NP1 on the T1D risk did not replicate in the full DAISY cohort. One explanation may be limited power. There was a smaller proportion of cases of T1D to people without T1D in the cohort. NP1 may have a small impact on the oxylipin profile and, thus, on the risk of T1D. A larger sample size may be needed to capture this small difference in risk. Additionally, genetic factors and environmental triggers may have larger and more lasting impacts on the oxylipin profile, and these environmental triggers instigating inflammation may be a more potent target for reducing T1D risk. The nested case-control study may not be representative of the full cohort. There were T1D cases that were included in the analysis in the DAISY cohort, but were not included in the case-control, because these participants did not have measured oxylipins. The background characteristics of these T1D cases may be different than those in the nested case-control study. The average age of the T1D onset of the cases that were in the cohort analysis (but not the case-control study) was 13.96 ± 7.25 compared to an average age of the T1D onset of 9.67 ± 4.49 in the nested case-control study. The pathophysiology of the development of T1D [95], as well as the serum vitamin D levels [96] and genetic risk factors [97,98] has been shown to be different in early-onset compared to late-onset T1D. 

Additionally, when developing the nutrient patterns in the case-control studies, we used the intercept as a summary measure, which did not incorporate the standard error of the summary measure, and a joint Cox PH model was used to test the nutrient pattern, which did incorporate the standard error. Incorporating this uncertainty when using the joint Cox PH model may have led to the inability to replicate the findings. Inflammation is also a dynamic process, as is the synthesis of oxylipins. Using the summary measures may not adequately capture these fluctuations in oxylipin synthesis in response to an inflammatory stimulus. We utilized all the dietary measures and did not restrict them by age, but there may also be a critical time window during which an oxylipin-related diet might be effective.

The strengths of this study include multiple measures of diets throughout childhood, as well as multiple measures of numerous oxylipins. An additional strength was the ability to adjust for the genetic influences on oxylipins and the oxylipin profile. The limitations include the small sample size in the nested case-control studies and a lack of the generalizability, given that DAISY is a cohort of children at an elevated risk of T1D compared to the general population, owing to a selection based on a genetic risk factor and family history of T1D.

## 5. Conclusions

To our knowledge, this is the first study to implement an RRR with oxylipin profiles as the response variables. We identified a nutrient pattern associated with an oxylipin profile previously implicated in the risk of T1D, characterized by a low intake of total sugars, vitamin C and iron, and a high intake of LA, anthocyanidins and sodium. This nutrient pattern exhibited a protective effect against T1D risk in the nested case-control study. However, it was not significantly associated with T1D when tested longitudinally in the full DAISY cohort. Future studies are needed to replicate the findings from the nested case-control study in a separate population as well as to investigate the dietary and other modifiable factors for oxylipin levels.

## Figures and Tables

**Figure 1 nutrients-15-00945-f001:**
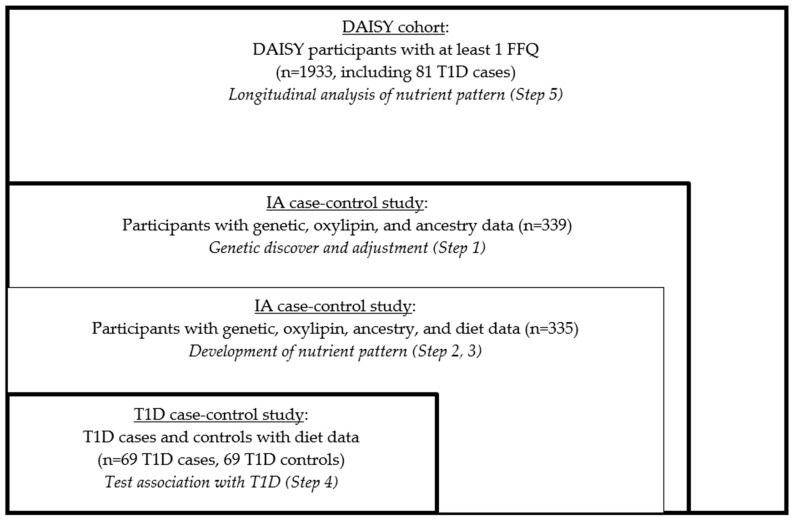
Study design and sample selection.

**Figure 2 nutrients-15-00945-f002:**
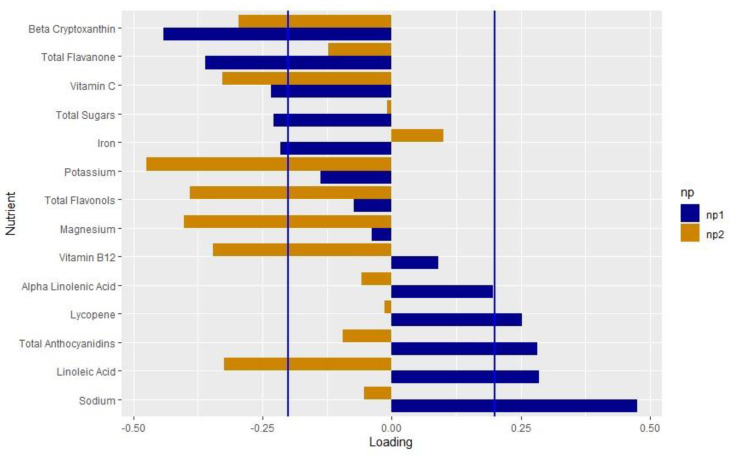
Loadings of the nutrient pattern: Loading values from the nutrient patterns extracted using reduced rank regression with genetically adjusted oxylipin PC1 and PC2 as the response variables. The nutrients are listed on the *y*-axis and the loading value is indicated on the *x*-axis. Blue bars indicate the loading values for NP1; orange indicates values for NP2. Blue lines mark the cutoff of the nutrients that load importantly, >|0.2|.

**Table 1 nutrients-15-00945-t001:** Participant characteristics: Type 1 Diabetes (T1D) nested case-control study.

Characteristic	T1D Cases (*n* = 69)	Controls (*n* = 69)	*p*-Value
Sex (female)	33 (47.8)	29 (42.0)	0.4936
Non-Hispanic White Ethnicity (yes)	61 (88.4)	63 (91.3)	0.5728
HLA–DR3/4 Genotype (yes)	34 (49.3)	13 (18.8)	0.0002
First Degree Relative with T1D (yes)	45 (65.2)	39 (56.5)	0.2953
Age at T1D Diagnosis	9.7 ± 4.5	n/a	

*n*(%) or mean ± SD. The high-risk T1D genotype Human Leukocyte Antigen (HLA) was defined as DRB1*04, DQB1*0302/DRB1*0301, DQB1*0201 (DR3/4 DQ8).

**Table 2 nutrients-15-00945-t002:** Individual associations between the nutrients and oxylipin PCs.

Univariate Association Between the Nutrients and Genetically Adjusted Oxylipin PC1		
**Nutrient**	**beta estimate**	** *p* ** **-Value**
Sodium	0.113	0.0014
Beta Cryptoxanthin	−0.106	0.0026
Total Flavanone	−0.086	0.0148
Total Anthocyanidins	0.068	0.0605
Linoleic Acid	0.068	0.0630
Lycopene	0.060	0.0922
Vitamin C	−0.058	0.1070
Total Sugars	−0.056	0.1274
Iron	−0.051	0.1526
Alpha Linolenic Acid	0.047	0.1933
**Univariate Association Between Nutrients and Genetically Adjusted Oxylipin PC2**		
**Nutrient**	**beta estimate**	** *p* ** **-value**
Potassium	−0.071	0.0534
Magnesium	−0.061	0.0979
Total Flavonols	−0.057	0.1094
Vitamin B12	−0.052	0.1477
Linoleic Acid	−0.052	0.1581
Vitamin C	−0.047	0.1948

Bivariate analysis of subject-specific calorie-adjusted nutrient intercept (exposure) and oxylipin PC1 and PC2 (outcome). Beta estimates are for a 1 standard deviation increase in the average calorie-adjusted nutrient intake.

**Table 3 nutrients-15-00945-t003:** Association of NP1 with T1D, adjusted for HLA, race/ethnicity, sex and family history of T1D in the nested case-control study.

Nutrient Pattern	OR	Lower CI	Upper CI	*p*-Value
NP1	0.442	0.233	0.840	0.0126
NP2	0.560	0.294	1.181	0.1362

Logistic regression model of NP1 and NP2 developed using reduced rank regression with the genetically adjusted oxylipin profiles as the response variables and tested in the nested case-control study. OR is for a 1 standard deviation increase in the nutrient pattern score. Upper and lower 85% confidence interval (CI) are displayed.

**Table 4 nutrients-15-00945-t004:** DAISY participant characteristics with at least one food frequency questionnaire (*n* = 1933).

Variable	Yes T1D (*n* = 81)	No T1D (*n* = 1852)	*p*-Value
Sex (female)	40 (49.4)	891 (48.1)	0.8225
Non-Hispanic White Ethnicity (yes)	74 (91.4)	1400 (75.6)	0.0011
HLA–DR3/4 Genotype (yes)	34 (41.2)	371 (20.0)	<0.0001
First Degree Relative with T1D (yes)	55 (67.9)	925 (50.0)	0.0016
Age at T1D Diagnosis	10.7 ± 5.4	n/a	

*n*(%) or mean ± SD. The high-risk T1D genotype (HLA) was defined as DRB1*04, DQB1*0302/DRB1*0301, DQB1*0201 (DR3/4 DQ8).

**Table 5 nutrients-15-00945-t005:** Longitudinal association of NP1 with T1D, adjusted for HLA, race/ethnicity, sex and family history of T1D.

Shared Parameter	HR	Lower CI	Upper CI	*p*-Value
Original NP1 (Average)	0.54	0.22	1.333	0.1829
Original NP1 (Cumulative)	0.98	0.84	1.129	0.7310
Simplified NP1 (Average)	0.52	0.22	1.240	0.1402
Simplified NP1 (Cumulative)	0.99	0.85	1.152	0.9054

The joint Cox PH model of NP1 developed using the reduced rank regression with the genetically adjusted oxylipin profiles as the response variables. HR is for a 1 standard deviation increase in the NP1 score. Upper and lower 85% confidence interval (CI) are displayed.

## Data Availability

The data presented in this study are available on request from the corresponding authors.

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
