# Peer review of "An Oxylipin-Related Nutrient Pattern and Risk of Type 1 Diabetes in the Diabetes Autoimmunity Study in the Young (DAISY)"

_nutrients, 2023, doi:10.3390/nu15040945_

Round 1

Reviewer 1 Report

The article and presentations are interesting. However,  I did not see was the BMI of the participants.  Higher BMI, indicating higher sugar, iron intake.... (ate too much of everything?). Not sure about the association of the higher VC and higher iron intake resulting in iron over loadings. Iron adsorption is little more complicated. Heme iron is 2-3 times more easily absorbed than fortified iron.  Inflammation reduce iron adsorption too. Maybe nutrient density (i.e., nutrition elements over total energy) as a measure instead of total nutrient intake? Can obesity be a trigger instead of particular nutrient or lack of a nutrient?

They need to at least address these issues.

Author Response

Thank you for the thoughtful and helpful review of our manuscript “An Oxylipin-Related Nutrient Pattern and Risk of Type 1 Diabetes in the Diabetes Autoimmunity Study in the Young (DAISY).” We list each comment below and our response follows, in italicized font. Additions are in red font.

Comments and Suggestions for Authors

The article and presentations are interesting. However, I did not see was the BMI of the participants. Higher BMI, indicating higher sugar, iron intake.... (ate too much of everything?).

-We thank the reviewer for this comment. We did not adjust for BMI in this analysis, since it is not associated with the outcome of type 1 diabetes in our population. In order to adjust for the potential of “eating too much of everything,” we residual-adjusted for calorie intake. This takes into account a participant who eats a high caloric amount, and therefore a higher amount of many nutrients.

Not sure about the association of the higher VC and higher iron intake resulting in iron over loadings. Iron adsorption is little more complicated. Heme iron is 2-3 times more easily absorbed than fortified iron.  Inflammation reduce iron adsorption too. Maybe nutrient density (i.e., nutrition elements over total energy) as a measure instead of total nutrient intake? Can obesity be a trigger instead of particular nutrient or lack of a nutrient?

They need to at least address these issues.

-Thank you for this comment. We agree that there are many factors that can impact both iron absorption and metabolism. The iron variables used in this analysis did not parse out heme and non-heme iron, and we also do not know whether vitamin C was eaten at a similar time as the vitamin C (and thus enhancing absorption). Additionally, we measured nutrient intake, not nutrient levels, and inflammation may impact nutrient absorption and levels. It will be important for future studies to unravel these complex relationships in the context of type 1 diabetes.

To address this, we added the following to the discussion (lines 351-356):

“In this analysis, we were not able to assess the difference between heme and non-heme iron, or the timing of iron and vitamin C intake, which may impact absorption levels. Future studies on the interplay between iron intake, vitamin C, and total sugars in the context of inflammation and T1D may reveal the mechanisms of these relationships in reducing risk of T1D.”

There are many methods of accounting for nutrient density, and we selected the residual method by Willett, since it is a common method in nutrition epidemiology for adjusting for energy intake. Although obesity and inflammation are closely linked, for T1D, obesity is not generally considered a trigger of islet autoimmunity (nor T1D in our population).

Reviewer 2 Report

In this nested case-control study of type 1 diabetes children, the authors created nutrient patterns using reduced rank regression by maximumly explaining the variation of two major blood oxylipins profiles that have been reported to be associated with type 1 diabetes in the same study population in their previously published paper. The first nutrient pattern extracted was associated with type 1 diabetes in case-control dataset set, but not in their original cohort where case and control were drawn from. The manuscript has been well written. I only have a few suggestions for potential improvement.

1. Page 5, 3) Nutrient Pattern Development

“We selected all nutrients that were associated with genetically-adjusted oxylipin PC1 or PC2 (p<0.2) using linear regression to be entered RRR. All nutrients associated with genetically-adjusted oxylipin PC1 and/or PC2 were used as the diet variables, and both genetically-adjusted oxylipin PC1 and PC” were used as the response variables for the RRR model. The nutrient patterns that best explain oxylipin variation were chosen by minimizing the predicted residual sum of square(PRESS). Nutrient loadings were used to interpret the nutrient patterns, and nutrients with loadings >|0.2|”were of interest.

 Does “nutrients” refer to “age-adjusted measure of average nutrient intake”?

 What is the reason the authors did not include all nutrient variables into RRR and then to see what variables having greater loading score? I think this procedure is more typical, simple and would be more interpretable especially considering RRR can handle a large amount of predictor variables. I am also curious about the results without pre-selection of variables. 

 2. Page 6, 5) Nutrient patterns and risk of T1D longitudinally within the full DAISY cohort.

“We applied the nutrient patterns to all FFQs measures by scoring each food record according to the nutrient pattern.”

How the authors scored food record according to nutrient pattern is not clear.

3. Page 8, 3.3 Nested Case-Control Association with T1DM

“NP1 was associated with a significantly lower risk of T1D (OR: 0.442, p=0.0126). “

I can not find this important result in any table or graph. 

Author Response

Thank you for the thoughtful and helpful review of our manuscript “An Oxylipin-Related Nutrient Pattern and Risk of Type 1 Diabetes in the Diabetes Autoimmunity Study in the Young (DAISY).” We list each comment below and our response follows, in italicized font. Additions are in red font.

Comments and Suggestions for Authors

In this nested case-control study of type 1 diabetes children, the authors created nutrient patterns using reduced rank regression by maximumly explaining the variation of two major blood oxylipins profiles that have been reported to be associated with type 1 diabetes in the same study population in their previously published paper. The first nutrient pattern extracted was associated with type 1 diabetes in case-control dataset set, but not in their original cohort where case and control were drawn from. The manuscript has been well written. I only have a few suggestions for potential improvement.

  1. Page 5, 3) Nutrient Pattern Development

“We selected all nutrients that were associated with genetically-adjusted oxylipin PC1 or PC2 (p<0.2) using linear regression to be entered RRR. All nutrients associated with genetically-adjusted oxylipin PC1 and/or PC2 were used as the diet variables, and both genetically-adjusted oxylipin PC1 and PC” were used as the response variables for the RRR model. The nutrient patterns that best explain oxylipin variation were chosen by minimizing the predicted residual sum of square(PRESS). Nutrient loadings were used to interpret the nutrient patterns, and nutrients with loadings >|0.2|”were of interest.

 Does “nutrients” refer to “age-adjusted measure of average nutrient intake”?

 What is the reason the authors did not include all nutrient variables into RRR and then to see what variables having greater loading score? I think this procedure is more typical, simple and would be more interpretable especially considering RRR can handle a large amount of predictor variables. I am also curious about the results without pre-selection of variables. 

-Thank you for this clarification question. The nutrient intercepts were age-adjusted. We have changed the wording to “We selected all nutrients (using the age-adjusted average intake measures) that were associated with genetically-adjusted oxylipin PC1 or PC2 (p<0.2) using linear regression to be entered in the RRR” for clarity (lines 224-227).

We agree that many researchers include all diet variables in reduced rank regression. In our study, due to the smaller sample size for the creation of the nutrient pattern, the reduced rank regression model did not produce nutrient patterns when all nutrients were included. Others have used a pre-selection process when creating dietary patterns (references below), so we implemented a similar method. We believe that this also directed the nutrient pattern to variables that were most closely related to oxylipin profiles as well.

We added the following references of studies that used a pre-selection process for the diet variables in a reduced rank regression (lines 224-227):

We selected all nutrients (using the age-adjusted average intake measures) that were associated with genetically-adjusted oxylipin PC1 or PC2 (p<0.2) using linear regression to be entered  in the RRR54-59.

  1. Schulze MB, Hoffmann K, Manson JE, et al. Dietary pattern, inflammation, and incidence of type 2 diabetes in women. The American journal of clinical nutrition 2005;82(3):675-84; quiz 714-5. doi: 10.1093/ajcn.82.3.675 [published Online First: 2005/09/13]
  2. Johnson RK, Vanderlinden LA, DeFelice BC, et al. Metabolomics-related nutrient patterns at seroconversion and risk of progression to type 1 diabetes. Pediatric diabetes 2020;21(7):1202-09. doi: 10.1111/pedi.13085 [published Online First: 2020/07/21]
  3. Hong JY, Kim YM, Shin MH, et al. Development and validation of dietary atherogenic index using common carotid artery-intima-media thickness: A food frequency questionnaire-based longitudinal study in Korean adults. Nutr Res 2022;104:55-65. doi: 10.1016/j.nutres.2022.04.006 [published Online First: 2022/05/27]
  4. Gu Y, Manly JJ, Mayeux RP, et al. An Inflammation-related Nutrient Pattern is Associated with Both Brain and Cognitive Measures in a Multiethnic Elderly Population. Curr Alzheimer Res 2018;15(5):493-501. doi: 10.2174/1567205015666180101145619 [published Online First: 2018/01/05]
  5. Jaacks LM, Crandell J, Mendez MA, et al. Dietary patterns associated with HbA1c and LDL cholesterol among individuals with type 1 diabetes in China. Journal of diabetes and its complications 2015;29(3):343-9. doi: 10.1016/j.jdiacomp.2014.12.014 [published Online First: 2015/01/30]
  6. Liese AD, Nichols M, Hodo D, et al. Food intake patterns associated with carotid artery atherosclerosis in the Insulin Resistance Atherosclerosis Study. The British journal of nutrition 2010;103(10):1471-9. doi: 10.1017/s0007114509993369 [published Online First: 2010/01/23]

  1. Page 6, 5) Nutrient patterns and risk of T1D longitudinally within the full DAISY cohort.

“We applied the nutrient patterns to all FFQs measures by scoring each food record according to the nutrient pattern.”

How the authors scored food record according to nutrient pattern is not clear.

-Thank you for this clarification. We have changed the wording to “We applied the nutrient patterns to all FFQs measures by scoring the nutrients calculated from each food record according to weights derived from the nutrient pattern60” (lines 254-255)

  1. Seah JYH, Ong CN, Koh WP, et al. A Dietary Pattern Derived from Reduced Rank Regression and Fatty Acid Biomarkers Is Associated with Lower Risk of Type 2 Diabetes and Coronary Artery Disease in Chinese Adults. The Journal of nutrition 2019;149(11):2001-10. doi: 10.1093/jn/nxz164 [published Online First: 2019/08/07]

  1. Page 8, 3.3 Nested Case-Control Association with T1DM

“NP1 was associated with a significantly lower risk of T1D (OR: 0.442, p=0.0126). “

I can not find this important result in any table or graph. 

-We have added a table of these results (Table 3, lines 315-316)

Round 2

Reviewer 1 Report

no comments anymore.